# Obstacle Circumvention Strategies During Omnidirectional Treadmill Walking in Virtual Reality

**DOI:** 10.3390/s25061667

**Published:** 2025-03-08

**Authors:** Marco A. Bühler, Anouk Lamontagne

**Affiliations:** 1School of Physical and Occupational Therapy, McGill University, Montreal, QC H3G 1Y5, Canada; marco.buhler@mail.mcgill.ca; 2Jewish Rehabilitation Hospital—CISSS Laval Site of the Centre for Interdisciplinary Research in Rehabilitation of Greater Montreal (CRIR), Laval, QC H7V 1R2, Canada

**Keywords:** gait, rehabilitation, virtual reality, omnidirectional treadmill, obstacle avoidance

## Abstract

Obstacle circumvention is an important task for community ambulation that is challenging to replicate in research and clinical environments. Omnidirectional treadmills combined with virtual reality (ODT-VR) offer a promising solution, allowing users to change walking direction and speed while walking in large, simulated environments. However, the extent to which such a setup yields circumvention strategies representative of overground walking in the real world (OVG-RW) remains to be determined. This study examined obstacle circumvention strategies in ODT-VR versus OVG-RW and measured how they changed with practice. Fifteen healthy young individuals walked while avoiding an interferer, performing four consecutive blocks of trials per condition. Distance at onset trajectory deviation, minimum distance from the interferer, and walking speed were compared across conditions and blocks. In ODT-VR, larger clearances and slower walking speeds were observed. In contrast, onset distances and proportions of right-side circumvention were similar between conditions. Walking speed increased from the first to the second block exclusively. Results suggest the use of a cautious locomotor behavior while using the ODT-VR setup, with some key features of circumvention strategies being preserved. Although ODT-VR setups offer exciting prospects for research and clinical applications, consideration should be given to the generalizability of findings to the real world.

## 1. Introduction

Navigating community environments requires pedestrians to avoid collisions with surrounding obstacles [1]. Unfortunately, successful obstacle circumvention, which relies on timely adaptations in both walking trajectory and speed [2,3], is often compromised in individuals with neurological conditions such as stroke [4] and Parkinson’s Disease [5], making the practice of such a task an essential component of rehabilitation. Doing so, however, poses challenges related to space, safety, and adaptation of the task according to patient abilities. Virtual reality (VR) offers a promising solution, allowing the immersion of patients in safe and ecological environments while providing full control of task parameters (e.g., speed, direction of approach, etc.). Another great advantage of VR is that it allows clinicians to obtain information about a patient’s abilities from the optical sensors and inertial measurement units that current systems use to track the user’s position in the virtual environment. However, a main drawback of VR is that it involves a medium to display visual information (VR headset or projection screen) and a locomotor setup (overground or treadmill) that imposes limitations in one way or another, resulting in VR experiences restricted to short overground walking distances or to walking on a fixed-speed, unidirectional conventional treadmill. Omnidirectional treadmills (ODTs) can address such limitations by enabling both trajectory and speed changes. Considering that ODTs can track the user’s movement, mainly via optical and/or IMU sensors, they can be used in combination with VR to provide a locomotor experience that mimics the real world (RW). This prospect is especially relevant to clinical settings where physical spaces are often small and crowded.

Currently available ODTs come in different shapes, sizes, and hardware implementations. The most popular implementation is non-motorized and involves a surface where the user’s feet slide with special footwear. This relatively low-cost implementation was, however, shown to involve greater energy expenditure and to result in substantial kinetic and kinematic changes compared to overground walking [6,7]. Recently, an ODT with a motorized “moving floor” design, the Infinadeck™, was released and promised a more natural walking experience. Using belts that can move forward and sideways, this treadmill allows free changes in speed and direction within a virtual environment. The usefulness of such ODT-VR setup in rehabilitation, however, is constrained by the extent to which it can elicit a natural locomotor behavior. Recently, a study reported slower walking speeds and greater step length variability during walking and turning in VR with the Infinadeck, compared to overground without VR [8]. Evidence from overground locomotor studies further suggests that VR itself, in comparison to the RW, induces conservative circumvention strategies in response to static or moving objects/pedestrians [9,10,11,12,13], as shown by slower walking speeds and larger obstacle clearances, while overall strategies (e.g., side of circumvention, adaptation to obstacle characteristics) remain preserved [9,10,11,12,13]. Whether the same would apply to a setup comprising a motorized ODT and VR, however, remains to be established.

With the long-term goal of using an ODT-VR setup to evaluate and train patients to perform complex locomotor tasks within virtual community environments, the present study aimed to estimate the extent to which obstacle circumvention strategies in response to a stationary interferer differ between ODT-VR and overground walking in the RW (OVG-RW). It further aimed to estimate the extent of adaptations occurring due to repeated practice in the ODT-VR condition. We hypothesized that due to the unfamiliarity of ODT walking and the known influence of VR on locomotion, participants would maintain larger clearances around the interferer and walk at slower speeds in the ODT-VR condition. In line with previous evidence [9], we further hypothesized that repeated practice with the ODT-VR setup would induce a cumulative increase in walking speed while leaving interferer clearance unchanged.

## 2. Materials and Methods

### 2.1. Participants

A sample of convenience of 15 healthy young participants was recruited. This sample size was selected based on an a priori power analysis that consisted of a significance level of 0.05, a power of 80%, two predictors, and based on similar studies and outcomes displaying a large effect size [6,8,14]. Inclusion criteria included the following: age between 18 and 29 years, no medical condition interfering with walking, and normal or corrected-to-normal visual acuity (log MAR 0.0 or better on the EDTRS [15]).

The study was approved by the Research Ethics Board in Rehabilitation and Physical Disability of the Centre for Interdisciplinary Research in Rehabilitation of Greater Montreal. Participants gave their written consent before being included in the study.

### 2.2. Experimental Procedure and Setup

Data were collected at the Virtual Reality & Mobility Lab at the Jewish Rehabilitation Hospital. Initially, participants underwent a clinical assessment. At this point, their body mass and height were measured using a weight scale and metric tape, their comfortable walking speed was obtained using the 10 m walk test, and their handedness was assessed with the Edinburgh Handedness Inventory [16]. Descriptive statistics for the clinical assessment are presented in Table 1.

In sequence, participants performed the experimental task. The task involved circumventing a static interferer in two conditions: ODT-VR and OVG-RW. The sequence in which these conditions were presented was randomized. In each condition, participants were randomly exposed to three trial types. Two of these conditions involved circumventing an interferer standing at 3.0 or 3.5 m from the participant’s start position. The third involved walking straight without an interferer. The layout of this task is depicted in Figure 1A.

In total, participants completed 24 trials. These trials were divided into four blocks, each containing two trials of each condition. Before data collection, participants were instructed to walk toward a target displayed on a TV screen after hearing a “Start” sound. In case the interferer was on their path, they should continue walking toward the target and circumvent the interferer in any way they deemed necessary. The Unity engine 2019 (Unity Software, San Francisco, CA, USA) was used to control experimental conditions, manage randomization, data recording, and display the virtual environment.

The OVG-RW condition took place in a large open room, with two HTC VIVE base stations 2.0 (HTC corporation, Taoyuan, Taiwan) positioned diagonally on opposite sides to cover a 7.5 × 5 m volume. Throughout the experiment, a male collaborator (height: 1.74 m; weight: 87 kg; age: 28) acted as an interferer. Both the participant and the interferer wore an HTC VIVE controller™ (HTC corporation, Taoyuan, Taiwan) on their sternum to track their position. This body landmark was chosen to maximize visibility by the base stations. For each participant, the distance from the controller to the participant’s body center along the anteroposterior axis was measured to offset the controller’s position data in Unity. Another collaborator, standing behind the participant, used hand signals to communicate the trial type to the interferer, indicating 3.0 m, 3.5 m, or no-obstacle trials. Participants closed their eyes while the interferer switched positions.

In the ODT-VR condition, participants walked on the Infinadeck™ (Rocklin, CA, USA) (Figure 1B) while viewing the virtual environment through an HTC VIVE Pro Eye™ headset (110° field of view, 1440 × 1600 pixels per eye, 90 Hz refresh rate). The virtual environment, created in Maya LT™ (Autodesk, San Francisco, CA, USA), matched the appearance and dimensions of the laboratory room used in the OVG-RW condition (Figure 1C). Additionally, the virtual agent in this environment replicated the interferer’s physical appearance and dimensions. Participant data were collected via an HTC VIVE tracker™ (HTC corporation, Taoyuan, Taiwan) attached to the participant’s lower back, which is also used to control the Infinadeck™. The trackers’ position was recorded using three HTC VIVE base stations positioned around the treadmill along an equilateral pattern. To calculate measurements, we recorded data from Infinadeck’s proprietary code, which updated the “player” position in the virtual environment based on ODT movements. Data from the virtual agent were tracked using its frontal chest landmark. If a collision with the interferer occurred, an “Ouch” sound was played and the trial ended.

For balance and comfort, participants were instructed to lightly touch the handrail of the Infinadeck while walking, a procedure implemented after pilot testing. Before the ODT-VR condition, participants underwent a habituation session on the Infinadeck, which included walking straight and practicing turning. This habituation was performed first without VR (~2 min) and then with VR (until reaching comfortable overground walking speed for 1 min). To avoid habituation to the virtual environment used in the experiment, habituation took place in a virtual shopping mall. After completing the ODT-VR condition, participants were questioned on their previous use of VR and answered the Simulator Sickness Questionnaire (SSQ) [17] to report on cybersickness symptoms.

### 2.3. Outcome Measures

Outcome measures were computed in MATLAB^®^ (version 9.14) using the HTC VIVE controller and tracker data, as detailed below. Walking trajectory adaptations were quantified using *Minimum distance*, *Maximum deviation*, *Onset distance*, and *Side of circumvention*. To calculate the *Minimum distance*, also referred to as obstacle clearance, we computed the Euclidean distance between the participant and the obstacle and then identified the smallest value. *Maximum deviation* was defined as the peak mediolateral displacement. The *Onset distance* was the Euclidean distance between the participant and the obstacle at the point at which participants initiated a trajectory deviation. This onset of trajectory deviation was identified using a method described in our previous work [18]; briefly, it was defined as the first instance of zero mediolateral velocity that was followed by a mediolateral displacement greater than that observed in unobstructed trials. Finally, the *Side of circumvention* was identified.

Walking speed adaptations were obtained by taking the first derivative of the participant’s walking trajectory and then extracting their *Minimum*, *Average*, and *Maximum* values, within a time window that began after the participant’s initial acceleration and ended at the point of interferer crossing. Initial acceleration was defined as the first frame after walking began in which acceleration in the anteroposterior direction was ≤0 for at least 0.25 s.

### 2.4. Statistical Analysis

Generalized estimating equations (GEE) were used to examine changes in outcome measures due to two within-subject factors: condition (ODT-VR vs. OVG-RW) and block (blocks 1–4). The 3.0 m and 3.5 m trial types were combined to describe the participant’s overall circumvention strategies and unobstructed trials were used to contrast the participant’s straight walking speed. No collisions occurred in either condition. However, four trials were excluded from the analysis due to a technical issue in the OVG-RW condition. Within-subject correlations were modeled using an exchangeable correlation structure. The significance level was set to *p* < 0.05 and Bonferroni corrections were applied for multiple comparisons. Analyses were conducted using SAS 9.4.

## 3. Results

Four participants had never used VR, while seven reported using it once or twice. The remaining four participants indicated monthly (n = 2), weekly (n = 1), or daily (n = 1) VR use. All were able to reach their overground walking speed during habituation. Participants’ average Simulator Sickness Questionnaire (SSQ) scores were as follows: Total score = 16.95, Nausea score = 19.08, Oculomotor score = 12.13, and Disorientation score = 13.92.

Figure 2 depicts participants’ trajectories and walking speed profiles in the ODT-VR and OVG-RW conditions, while trajectory adaptation outcomes are displayed in Figure 3. Statistical analyses revealed that minimum distance (X^2^ (1, 476) = 7.96, *p* < 0.01) and maximum deviation (X^2^ (1, 476) = 9.38, *p* < 0.01) were significantly larger in the ODT-VR condition, reaching a difference of 0.19 and 0.18 m, respectively. A trend toward smaller onset distances (i.e., 0.13 m) in the ODT-VR condition was observed but it did not reach statistical significance (X^2^ (1, 476) = 3.80, *p* = 0.05). Percentages of right circumventions in the OVG-RW (62.7%) and the OVG-VR (65%) conditions were qualitatively similar. As for the effect of block (i.e., repeated practice) or the interaction between condition and block, no statistically significant differences were observed for minimum distance, maximum deviation, or onset distance.

Concerning walking speed adaptations, a significant effect of condition was observed for minimum (X^2^(1, 476) = 13.38, *p* < 0.001), average (X^2^(1, 476) = 12.25, *p* < 0.001), and maximum (X^2^(1, 476) = 10.63, *p* < 0.001) walking speeds. Respectively, walking speeds were 0.40, 0.41, and 0.34 m/s slower in the ODT-VR condition (Figure 4). A similar decrease was observed in the ODT-VR condition for unobstructed trials (Δ = 0.41 m/s).

In contrast to measures of trajectory adaptation, a significant effect of block was observed for minimum (X^2^ (3, 476) = 11.16, *p* < 0.05), average (X^2^ (1, 476) = 11.08, *p* < 0.05), and maximum (X^2^ (1, 476) = 10.63, *p* < 0.05) walking speeds. Post hoc comparisons revealed a 0.07 m/s (*p* < 0.001) increase in minimum, average, and maximum walking speeds from the first to the second block. For unobstructed trials, a similar increase in walking speed (Δ = 0.07 m/s, *p* < 0.001) was observed between the first and second block only. No statistically significant interactions between condition and block were observed for speed-related outcomes.

## 4. Discussion

This study compared, for the first time, obstacle circumvention when walking on a motorized omnidirectional treadmill in VR (ODT-VR) versus overground in the real world (OVG-RW). Our results showed that ODT-VR led to larger trajectory deviations and obstacle clearances, as well as slower walking speeds, which aligns with previous studies on obstacle circumvention in VR [9,10,11,12,13] and ODT walking in VR [14] and suggests a more cautious obstacle circumvention behavior. Such differences may have arisen, at least in part, from a distance perception bias causing users to perceive virtual objects as being closer than their actual distance [19], resulting in larger trajectory adaptations as well as slower walking speeds [9,10,11,13,18]. Such perception is thought to result from the reduced field of view and availability of depth perception cues in VR, as well as from poor knowledge of object size in VR, and would be aggravated by headset weight and a lower sense of presence [19,20,21,22,23,24]. Magnitude-wise, however, discrepancies in obstacle clearance (∆ = 0.18–0.19 m) observed here are larger than in overground walking VR studies (Δ = 0.06–0.16 m) [9,10,11,12,13]. Similarly, the observed walking speed reduction (∆ = 0.41 m/s) in the ODT-VR condition exceeds those reported for a similar obstacle circumvention task performed overground in VR (∆ = 0.04–0.17 m/s) [9,10,11,12,13]. These observations suggest that the treadmill itself also affects locomotor behavior. This ‘treadmill effect’ likely results from the novelty of ODT walking and from factors associated with the treadmill control [6,8]. Interestingly, observed differences in obstacle clearance between ODT-VR and OVG-RW remain much smaller (≈50%) than that reported while using a Virtuix VR Omni treadmill (∆ = 0.42 m) [14], a non-motorized treadmill that uses a low-friction walking surface, suggesting that the Infinadeck’s “moving floor” system allowed for a more naturalistic and effective navigation around virtual obstacles.

As for alterations in walking speed, a recent study using the Infinadeck reported a dramatic reduction in walking speed during unobstructed walking (Δ = 0.80 m/s) [8]. While habituation with the treadmill was also provided in that study, it was considered complete when the participant reported feeling comfortable, and the gait was deemed consistent by an observer. In the present study, participants were instead habituated until reaching their comfortable walking speed overground, highlighting the importance of a potentially longer habituation for an optimal use of the ODT-VR setup. However, participants’ walking speed on the Infinadeck remained slower than during OVG-RW, for both obstructed and unobstructed walking. While this suggests that the slower speeds are attributable to the ODR-VR setup rather than a circumvention strategy, we cannot rule out the possibility that during their initial interactions with the interferer, participants selected a preferred walking speed for the task and maintained this speed consistently across both obstructed and unobstructed trials.

Present results also revealed similar distances at onset of trajectory deviation and preferred side of circumvention between the ODT-VR and OVG-RW conditions. Such similarities are consistent with previous circumvention studies that involved overground walking in VR versus RW [9,10]. Considering that obstacle circumvention follows a two-phase process, which is an anticipatory phase where circumvention is initiated by small adjustments made far from the obstacle, and a clearance phase where larger adjustments ensure safe obstacle clearance [25,26], we suggest that adjustments in the anticipatory phase are less impacted by VR. Supporting this assertion, distance perception for distant objects was shown to be less affected by VR [19,27], which might have helped users make anticipatory locomotor adaptations consistent with the OVG-RW condition. Additionally, the slow walking speed of participants in this condition, by providing more time to evaluate one’s distance from the interferer, may have helped preserve similar onset distances of trajectory deviation as in the RW. Collectively, these findings suggest that differences brought by ODT-VR manifest in the magnitude of certain variables rather than changes in the fundamental characteristics of circumvention strategies.

Results also show that participants maintained constant trajectory adaptations with repeated task exposure, which aligns with our previous study on pedestrian circumvention in VR versus RW during overground walking [9]. While repeated walking in a virtual environment triggers a perception recalibration that improves egocentric distance perception [23], both this study and our previous one involving overground walking [9] used a virtual environment replicating the real laboratory. Participants’ prior spatial knowledge of the environment likely enhanced distance perception in VR [19,28], thereby reducing the need for recalibration and resulting in consistent performance throughout the experiment. Exclusively between the first and second block, repeated practice also led to a small increase in walking speed (0.07 m/s). As this was observed in both ODT-VR and OVG-RW, it suggests a similar walking speed adaptation between the two conditions. While this increase in walking speed was indeed statistically significant, its magnitude does not reach the threshold for a minimal clinically important difference (0.1 m/s) [29]. This partially aligns with our earlier work involving overground walking in VR and RW [9], where speed increments were observed throughout the experiment. Unlike this earlier work, however, this study included a habituation period that might have masked the initial speed adaptations. Furthermore, following this habituation and 24 experimental trials, walking speed remained slower compared to OVG-RW, underscoring the need for further improvements to ODT-VR technologies.

### Limitations

Participants were instructed to lightly touch the rail during ODT-VR walking, providing kinesthetic feedback that likely aided balance and prevented a further reduction in walking speed [8]. Thus, our findings may not fully apply to ODT-VR without handrail use. Additionally, comparing ODT-VR to OVG-RW limited our ability to isolate the respective effects of the ODT and VR on circumvention strategies. While such insights would have been valuable, our primary goal was to assess how effectively the ODT-VR setup we propose for clinical applications can replicate real-world pedestrian interactions. Finally, findings should be interpreted considering participant demographics, the VR headset and the ODT used; caution should be used when generalizing to other populations (e.g., older adults, individuals with disabilities) or ODT-VR setups.

## 5. Conclusions

This study examined circumvention strategies in response to a static interferer during ODT-VR and OVG-RW walking. Results revealed larger obstacle clearances and slower walking speeds in ODT-VR but similar onset distances of trajectory deviation and preferred side of circumvention compared to OVG-RW. While some differences may stem from VR, ODT control mechanisms, and participants’ familiarity with the setup, the observed variations are expressed in magnitude rather than changes in circumvention strategies. Repeated exposure mildly increased walking speed early in the task for both conditions, with other variables remaining unchanged. Setups that integrate ODT and VR, like the one examined here, hold great promise for gait rehabilitation. These systems enable clinicians to assess and train one’s ability to perform complex locomotor tasks in conditions that resemble real-life situations within a safe and controlled environment. However, when interpreting results or designing assessments and interventions, it is essential to carefully consider the effects of ODT-VR systems on locomotor behaviors.

## Figures and Tables

**Figure 1 sensors-25-01667-f001:**
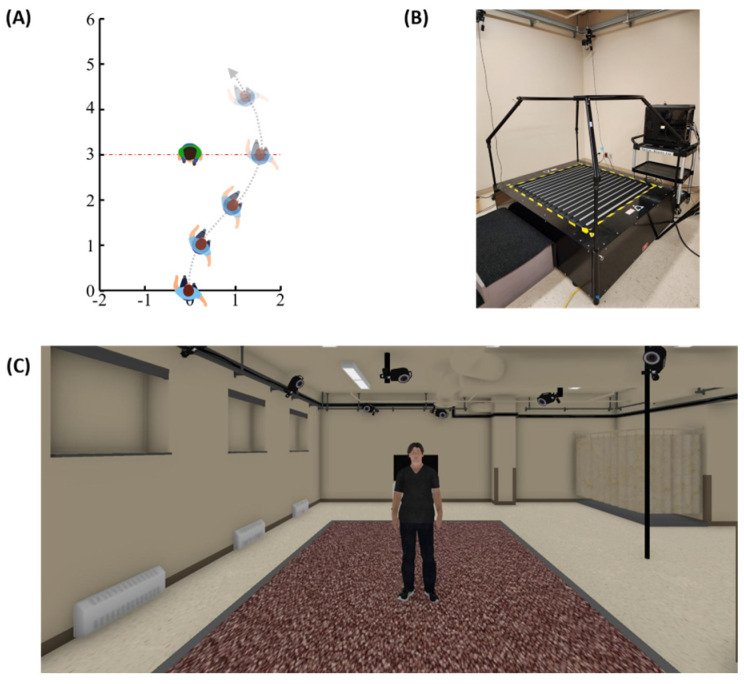
The layout of the obstacle circumvention task from a bird’s eye view with the interferer standing at 3 m from the start position (**A**). Photograph of the Infinadeck (**B**). Virtual environment depicting a condition with the interferer standing at 3m from the start position (**C**).

**Figure 2 sensors-25-01667-f002:**
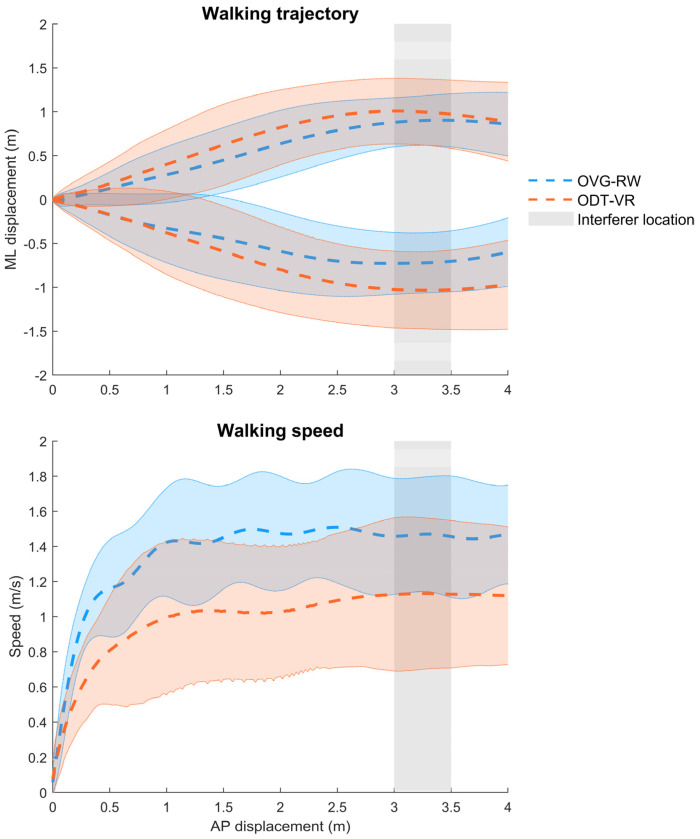
Mean ± 2SD Traces representing walking trajectories (Top) and walking speed (Bottom) from all participants in both the OVG-RW (Blue) and ODT-VR (Orange) conditions. Shaded gray area marks the positions where the interferer would be standing (3.0 or 3.5 m).

**Figure 3 sensors-25-01667-f003:**
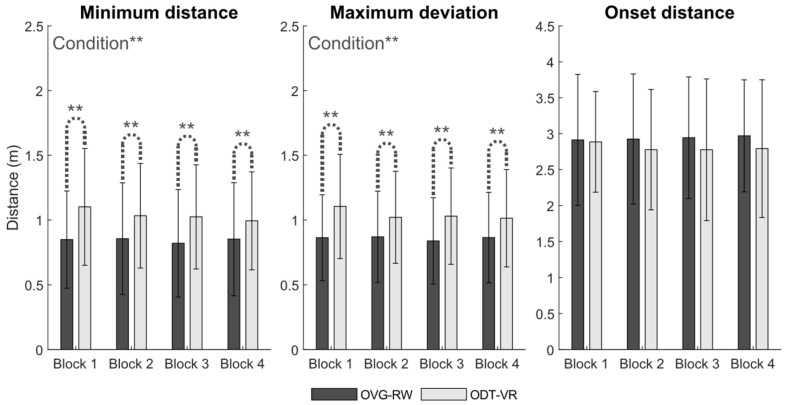
Mean ± 2SD values of all participants for Minimum distance, Maximum deviation, and Onset distance. ** *p* < 0.01.

**Figure 4 sensors-25-01667-f004:**
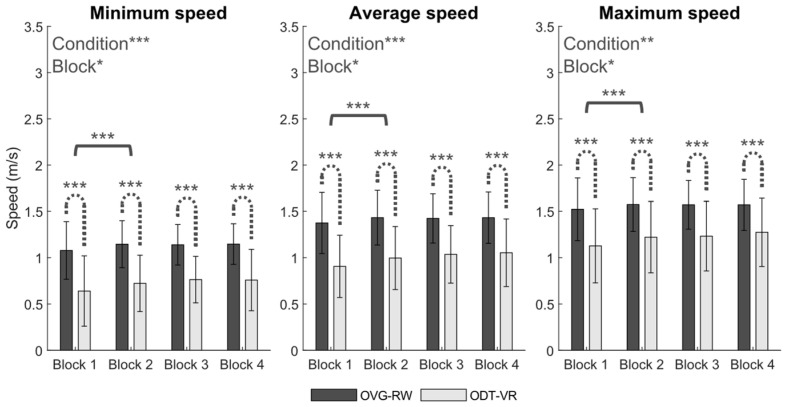
Mean ± 2SD values of all participants for Minimum, Average, and Maximum speed. * *p* < 0.05, ** *p* < 0.01, *** *p* < 0.001.

**Table 1 sensors-25-01667-t001:** Sample Characteristics.

	Mean	SD
Age (years)	23.8	3.21
Height (m)	1.69	0.09
Mass (Kg)	67.41	12.7
Comfortable walking speed (m/s)	1.46	0.19
	Male	Female
Sex	8	7
	Right	Left
Handedness	13	2

## Data Availability

The original data presented in this study will be made openly available in the McGill Dataverse repository https://doi.org/10.5683/SP3/HKWJ2Z (accessed on 28 January 2025).

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
