# Peer review of "Obstacle Circumvention Strategies During Omnidirectional Treadmill Walking in Virtual Reality"

_sensors, 2025, doi:10.3390/s25061667_

Round 1
Reviewer 1 Report
Comments and Suggestions for Authors
Authors present research results for obstacle circumvention strategies during omnidirectional treadmill walking in virtual reality (ODT-VR) and compares it with the overground walking in the real world (OVG-RW). The research is important for the gait rehabilitation, and usage of technology could significantly facilitate the exercises routine and provide insight in many parameters otherwise not accessible.
The structure of the manuscript is good, experimental study described in details, statistical methods adequate, and results and discussion provided in a clear and comprehensive manner.
The authors are precise and strict in noting all the limitations of this approach, and they note that "while the ODT-VR setup offers significant potential for research and clinical use, its effects on locomotor behaviors should be carefully considered when interpreting results or designing assessments and interventions." I would suggest that taking into account this reserve, it would be appropriate if authors includes the potential and benefits of employing VR assisted techniques for gait rehabilitatio,
Author Response
Comment 1: Authors present research results for obstacle circumvention strategies during omnidirectional treadmill walking in virtual reality (ODT-VR) and compares it with the overground walking in the real world (OVG-RW). The research is important for the gait rehabilitation, and usage of technology could significantly facilitate the exercises routine and provide insight in many parameters otherwise not accessible.
The structure of the manuscript is good, experimental study described in details, statistical methods adequate, and results and discussion provided in a clear and comprehensive manner.
The authors are precise and strict in noting all the limitations of this approach, and they note that "while the ODT-VR setup offers significant potential for research and clinical use, its effects on locomotor behaviors should be carefully considered when interpreting results or designing assessments and interventions." I would suggest that taking into account this reserve, it would be appropriate if authors includes the potential and benefits of employing VR assisted techniques for gait rehabilitation,
Response: First, we would like to thank the reviewer for their comments. We modified the last sentence of the conclusion section to address the last point mentioned above. Please see the changes below:
“Setups that integrate ODT and VR, like the one examined here, hold great promise for gait rehabilitation. These systems enable clinicians to assess and train one's ability to perform complex locomotor tasks in conditions that resemble real-life situations within a safe and controlled environment. However, when interpreting results or designing assessments and interventions, it is essential to carefully consider the effects of ODT-VR systems on locomotor behaviors.”
Reviewer 2 Report
Comments and Suggestions for Authors
The study provided more insights on obstacle circumvention strategies using omnidirectional treadmill and VR. Particularly on the observed variation of changes in variation magnitude rather than strategies of walking. well done.
Some suggestions
1) For the methodology section line 91 to 100, it might be clearer to the reader if there is a flow diagram to show the sequence of tests conducted as it is quite lengthy to read and understand the process with a single read.
2) Line 174-175, the shaded area in figure 2 should be the vertical bar on the graph. As there are multiple shades in the graph, it might be better to clearly state this in the figure itself.
3) Line 208, best not to start a sentence with an abbreviation "ODT-VR".
4) For figure 3 and figure 4 what is "*" representing. There are many "**,***" of different numbers which is confusing. Do add that in the table legend if necessary.
5) You have a total of 15 subjects in this study. Do provide some information to justify that there is sufficient power to detect a difference.
6) There is a difference of 0.07m/s changes in the post hoc analysis between block 1 and 2. (Line197-200). Statistically this is significant but I am not sure if this small value of change may translate to any clinical significant real world impact. Might be good to address this in your discussion or conclusion.
Author Response
Please see attached file that contains our responses and the modified figure.
